# Longitudinal changes in structural lung abnormalities using MDCT in chronic obstructive pulmonary disease with asthma-like features

**Rie Anazawa[1], Naoko Kawata[1]\*, Yukiko Matsuura[1], Jun Ikari[1], Yuji Tada[1], Masaki Suzuki[1], Shin Takayanagi[1], Shin Matsuoka[2], Shoichiro Matsushita[2], Koichiro Tatsumi[1]**

**1** Department of Respirology, Graduate School of Medicine, Chiba University, Inohana, Chuo-ku, Chiba-shi, Chiba, Japan, **2** Department of Radiology, St. Marianna University School of Medicine, Sugao, Miyamae-ku, Kawasaki-shi, Kanagawa, Japan

\* chumito_03@yahoo.co.jp

## Abstract

### Background

Some patients with chronic obstructive pulmonary disease (COPD) have asthma-like features. However, there have been few reports on the structural lung abnormalities found in this patient population. Multi-detector computed tomography (MDCT) can detect emphysematous low-attenuation areas (LAA) within the lung, airway thickness (wall area percentage, WA%), and the loss of pulmonary vasculature as the percentage of small pulmonary vessels with cross-sectional area (CSA) less than 5 mm$^2$ (%CSA<5). We analyzed differences in structural lung changes over time between patients with COPD and those with COPD with asthma-like features using these CT parameters.

### Material and methods

We performed pulmonary function tests (PFTs), MDCT, and a COPD assessment test (CAT) in 50 patients with COPD and 29 patients with COPD with asthma-like features at the time of enrollment and two years later. We analyzed changes in clinical parameters and CT indices over time and evaluated differences in structural changes between groups.

### Results

The CAT score and $FEV_1$ did not significantly change during the follow-up period in either group. Emphysematous LAA regions significantly increased in both groups. The %CSA<5 showed a small but significant increase in COPD patients, but a significant decrease in patients with COPD with asthma-like features. The WA% at the distal bronchi was significantly decreased in COPD, but did not significantly change in COPD with asthma -like features.

**Data Availability Statement:** All relevant data are within the paper and its Supporting Information files.

**Funding:** NK recieved the grants from the Ministry of Education, Science, Sports and Culture, Grant-in-Aid for Scientific Research (C) (16K01407,19K12816), the Chiba Foundation for Health Promotion & Disease Prevention(No.1272). KT recieved the grants from the Respiratory Failure Research Group (H26-Intractable diseases-General-076) from the Ministry of Health, Labour and Welfare, Japan. The funders had no role in study design, data collection and analysis, decision to publish, or preparation of the manuscript.

**Competing interests:** The authors have declared that no competing interests exist.

## Conclusion

Emphysematous LAA increased in patients with COPD with and without asthma-like features. The %CSA<5 and WA% at the distal bronchi did not change in parallel with LAA. Furthermore, changes in %CSA<5 were significantly different between patients with COPD and those with COPD with asthma-like features. Patients with COPD with asthma-like features may have different longitudinal structural changes than those seen in COPD patients.

## Background

Chronic obstructive pulmonary disease (COPD), a common disease that is increasing in prevalence, is characterized by persistent respiratory symptoms and chronic airflow limitation [1]. Worldwide, COPD is the third leading cause of death, and mortality continues to rise [2]. The chronic airflow limitation in COPD is caused by a mixture of small airway disease and parenchymal destruction, which leads to multiple COPD phenotypes. For example, the classical phenotype is emphysema/chronic bronchitis, and other phenotypes are frequent-exacerbator/non-exacerbator [3, 4]. Recently, "asthma- COPD overlap" (ACO) has been recognized in clinical practice. A joint project of the Global Initiative for Asthma (GINA) and the Global Initiative for Chronic Obstructive Lung Disease (GOLD) published a consensus on ACO [5]. The prevalence of ACO has been reported to be between 13% and 55% [6]. It has not been defined definitively, and previous studies have used multiple different diagnostic criteria. The asthmatic components for the diagnosis of ACO include history of asthma; asthma-like symptoms, positive bronchodilator test, blood eosinophilia, high total IgE, and/or atopy etc. [6–8]. GINA/GOLD applies a stepwise approach for the diagnosis of ACO, in which each characteristic of COPD and asthma are assessed. However, some patients with COPD have asthma-like features [9], even if they are not diagnosed as ACO. COPD with asthma-like features is a clinical phenotype of COPD and this phenotype is associated to individualized treatment [9–11].

Recently, multi-detector computed tomography (MDCT) has been used to evaluate structural changes in the lung. The loss of lung tissue associated with emphysema is detected as low-attenuation areas (LAA) on CT scans [12]. LAA percentage (LAA%) in patients with COPD is higher than in normal subjects [13], and correlates with COPD mortality independently of pulmonary function tests [14]. Airway wall thickening can also be measured using CT. Nakano et al. found that wall thickening in the apical bronchus of the right upper lobe correlated significantly with $FEV_1$ in patients with COPD [15]. The inner luminal area (Ai) and the wall thickness (wall area percentage, WA%) were measured at the third- to the sixth-generation airways. These indices correlated significantly with airflow limitation, and such correlations were more closely related to the distal than the proximal airways [16]. Moreover, chronic respiratory symptoms were positively associated with WA% in airways with a luminal diameter between 5 and 10 mm [17]. Recently, several studies have used a novel CT marker to evaluate the area of small pulmonary vessels, the percentage of small pulmonary vessels (%CSA). The %CSA<5 reflects pulmonary vascular alteration and correlates with airflow limitation, the extent of emphysema, and pulmonary hypertension in patients with COPD [18, 19].

Although there are many longitudinal studies on health status and lung function in COPD [20], longitudinal structural changes in COPD with asthma-like features have not been fully investigated. Several studies have been performed in patients with emphysema [21, 22], but there are few reports on changes in airway [23] or vascular remodeling [24]. The structural characteristics of ACO and COPD are different. Hardin et al. reported that patients with ACO

had less emphysema and greater airway wall thickness compared to patients with COPD [25]. We previously examined longitudinal structural changes in COPD patients and found that WA% at the distal bronchi and %CSA<5 did not change in parallel with LAA [26]. However, differences in longitudinal structural changes between the patients with COPD versus those with COPD with asthma-like features have not been fully elucidated.

The recognition of COPD with asthma-like features by the assessment of their structural characteristics may provide essential information for individualized management of patients with COPD. The present study aimed to investigate the longitudinal structural changes in patients with COPD with asthma-like features using MDCT. Such changes include changes in pulmonary emphysema, airway disease, and pulmonary vascular alteration. This study also aimed to evaluate the differences in structural abnormalities between COPD patients and patients with COPD with asthma-like features.

## Materials and methods

### Subjects

This prospective observational study enrolled 137 patients who presented to Chiba University Hospital from June 2010 to August 2016 for management of COPD. Patients were excluded if they had abnormal lung parenchymal lesions other than emphysematous change. At enrollment, 10 patients were excluded for the following reasons: 4 interstitial pneumonia, 1 infectious pneumonia, 1 old pulmonary tuberculosis, and 4 lung cancer or nodules. At the follow up period, 15 patients were excluded: 6 interstitial pneumonia, 2 infectious pneumonia, 3 lung cancer or nodules, 2 cardiac failure, and 2 pleural effusion. Finally, 50 patients with COPD and 29 patients with COPD with asthma-like features were enrolled in the study (Fig 1). The diagnosis of COPD was based on chronic respiratory symptoms, with smoking history, physical examination and results of spirometry according to the American Thoracic Society (ATS) and European Respiratory Society (ERS) recommendations [27]. Pulmonary function tests (PFTs) were performed with a Fudac-60 (Fukuda Denshi; Tokyo, Japan). Spirometric measurements

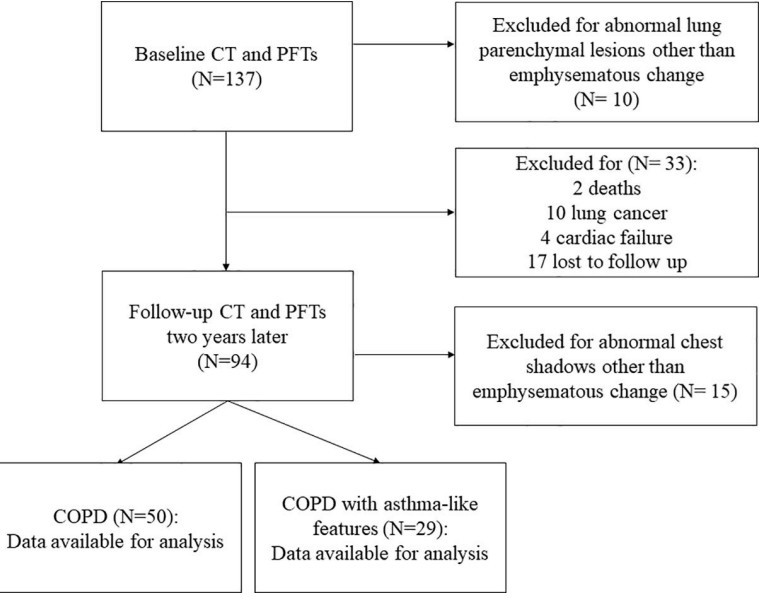

**Fig 1. Flow chart of the study participants.**

included forced vital capacity (FVC) and forced expiratory volume in 1 second ($FEV_1$), which were expressed as their predicted values based on the Japanese Respiratory Society (JRS) guidelines [28].

In this study, patients with COPD who experienced asthma symptoms (episodes of breathlessness, wheezing, cough, and chest tightness that were worse at night or in the early morning), bronchodilator reversibility (improvement greater than or equal to 200ml and 12% increase in $FEV_1$ compared to pre-bronchodilator data), blood eosinophils >5% or 300/ml, high levels of serum IgE, and/or history of allergic rhinitis were diagnosed with COPD with asthma-like features, in accordance with GINA and GOLD guidelines [5].

Patients underwent a COPD assessment test (CAT), PFTs and MDCT at the time of enrollment and two years later. Exacerbations were defined as a deterioration of respiratory symptoms that required antibiotics, systemic steroids and/or hospital admission [22].

The Ethics Committee of the Chiba University School of Medicine approved the study protocol (approval number: 857). Written informed consent was obtained from all study participants.

## MDCT scanning

All patients were scanned with a 64-MDCT scanner (Aquillion ONE, Toshiba Medical Systems; Tokyo, Japan) from the thoracic inlet to the diaphragm while at full inspiration. None received contrast medium. The scan was carried out with the following settings: 0.5-mm collimation; 120kV; auto-exposure control; gantry rotation time of 0.5 seconds; and beam pitch of 0.83. All images were reconstructed using standard reconstruction algorithms with a slice thickness of 0.5 mm and a reconstruction interval of 0.5 mm. The voxel size was 0.63×0.63×0.5 mm.

## MDCT measurements

For the measurements of LAA and CSA, we selected three CT slices, which were taken at 1 cm above the upper margin of the aortic arch (upper lung fields), 1 cm below the carina (middle lung fields), and 1 cm below the right inferior pulmonary vein (lower lung fields) [18]. These CT images were analyzed using ImageJ software, Version 1.44 (imagej.nih.gov/ij/download/).

LAA measurements were conducted as follows: The threshold technique for the total lung area (TLA) was adopted between -500 and -1024 Hounsfield units (HU). Since LAA was detected with an attenuation of -950 HU, images were converted into binary images with a window level of -950 HU. The range of circularity was set from 0 to infinity. With these settings, the LAA was automatically calculated. The LAAs of each lung field were summed, and the average of these values was determined.

CSA measurements were conducted as follows: The threshold technique for the total lung field was adopted between -500 and -1024 HU. Segmented images were converted into binary images with a window level of -720 HU. The range of circularity was set from 0.9 to 1.0 using the "analyze particles" function of the ImageJ software. With these settings, the CSA was measured separately by the size of each vessel (less than 5 $mm^2$; CSA<5). The CSAs of each lung field were summed, and the average of these values was determined. The percentage of CSA<5 (%CSA<5), and LAA (%LAA) in the TLA were calculated.

We used the free open-source software Airway Inspector (Brigham and Women's Hospital) for the measurements of WA%. WA% measurements were conducted according to the method of Yamashiro et al. [29]. We identified the third and fifth generations of the B1 and B10 bronchi in the right lung. The measured point was peripherally next to the branching points in each generation. Ai and the outer area of the bronchus (Ao) were measured

semiautomatically with the full-width at half-maximum method of the software. The WA% was defined with the equation WA% = 100 (Ao—Ai) / Ao. The MDCT measurements were evaluated independently by two pulmonologists (RA and YM). All data were anonymized and the observers were blinded to other characteristics of the subjects when the imaging analyses were performed.

## Statistical analysis

JMP 13.0 software was used for all statistical analyses (SAS Institute, Cary, NC), with the results expressed as mean (± SD) unless otherwise indicated. Comparisons of longitudinal changes in CT parameters, PFTs, and CAT scores between enrollment and follow-up in each group were performed using the Wilcoxon signed-rank test. Comparisons were performed with a chi-square test for categorical variables. Differences in the data between the patients with COPD and COPD with asthma-like features were performed with the Mann-Whitney $U$ test. For all statistical analyses, the level of significance was set at p <0.05.

## Results

### Patient characteristics

The clinical characteristics of the patients are presented in Table 1. There were 50 patients with COPD (44 men and 6 women; mean age, 70.1 ± 6.9 years) and 29 patients with COPD with asthma-like features (23 men and 6 women; mean age, 67.9 ± 8.5 years). There were no differences between the two groups in age, sex, percentage of ex/current smokers, smoking history or body mass index. All of the patients with COPD with asthma-like features met the criteria for asthma symptoms, whereas 9 (31%) had bronchodilator reversibility, 17 (59%) had blood eosinophilia, 13 (45%) had high levels of serum IgE, and 2 (7%) had a history of allergic rhinitis. The number of patients in each GOLD stage were as follows: COPD group: stage I, n = 13 (26%); stage II, n = 27 (54%); stage III, n = 7 (14%); stage IV, n = 3 (6%); COPD with asthma-like features group: stage I, n = 3 (10.3%); stage II, n = 12 (41.4%); stage III, n = 11 (38%); stage IV, n = 3 (10.3%). The COPD with asthma-like features group contained more patients with severe GOLD stage disease ($P$ = 0.048), and the frequency of exacerbation tended to be higher in the group with COPD with asthma-like features (COPD vs asthma-like features: 1.4 ± 1.0 vs 3.1 ± 3.8, $P$ = 0.0617). Additionally, the number of patients who received COPD treatment at enrollment and two years later was higher in the COPD with asthma-like features group (COPD vs asthma-like features: at enrollment 30% vs 89.7%, $P$ < 0.0001, 2 years later 64.0% vs 96.6%, $P$ = 0.0003).

### CAT score and PFTs

The CAT score, $FEV_1$, and $FEV_1$%predicted did not significantly change during the follow-up period in either group (Tables 2 and 3).

### CT parameters

The CT parameters measured at enrollment and follow-up CT scans are shown in Table 2, while the annual changes in CT measurements are shown in Table 3. Fig 2 shows the changes in radiological parameters over two years of follow- up. There was no significant difference in all the CT parameters including LAA%, CSA<5, and WA% at enrollment between the two groups. LAA and LAA% significantly increased in both groups during the follow-up period (COPD: baseline vs two years later: 6.8 ± 10.4% vs 8.8 ± 11.7%, $P$ < 0.0001. COPD with asthma-like features: baseline vs two years later: 10.8 ± 13.0% vs 14.1 ± 14.6%, $P$ < 0.0001).

**Table 1. Characteristics of patients with COPD with and without asthma-like features.**

| | COPD Mean (±SD) n = 50 | COPD with asthma-like features Mean (±SD) n = 29 | P value (between groups) |
|---|---|---|---|
| Age (years) | 70.1 ± 6.9 | 67.9 ± 8.5 | NS |
| Sex (male/female) | 44 (88%) / 6 (12%) | 23 (79.3%) / 6 (20.7%) | NS |
| Ex-smokers/Current smokers | 40 (80%) / 10 (20%) | 27 (93.1%) / 2 (6.9%) | NS |
| Smoking history (pack-years) | 56.0 ± 30.7 | 57.8 ± 44.5 | NS |
| BMI (kg/m$^2$) | 23.8 ± 3.0 | 22.7 ± 3.4 | NS |
| GOLD classification (I/II/III/IV) | 13 (26%) / 27 (54%) / 7 (14%) / 3 (6%) | 3 (10.3%) / 12 (41.4%) / 11 (38.0%) / 3 (10.3%) | 0.048 |
| Exacerbations (+/-) | 12 (24%) / 38 (76%) | 9 (31.0%) / 20 (69.0%) | NS |
| Exacerbations per 1 year | 0.8 ± 0.5 | 1.6 ± 1.9 | NS |
| COPD Treatment (Yes/No) at enrollment | 15 (30%) / 35 (70%) | 26 (89.7%) / 3 (10.3%) | < 0.0001 |
| LAMA | 15 (30%) | 16 (55.2%) | |
| LABA | 1 (2%) | 8 (27.6%) | |
| ICS | 0 (0%) | 1 (3.4%) | |
| ICS/LABA | 4 (8%) | 14 (48.3%) | |
| LAMA/LABA | 0 (0%) | 2 (6.9%) | |
| COPD Treatment (Yes/No) after 2 years | 32 (64%) / 18 (36%) | 28 (96.6%) / 1 (3.4%) | 0.0003 |
| LAMA | 27 (54%) | 20 (69.0%) | |
| LABA | 8 (16%) | 6 (20.7%) | |
| ICS | 0 (0%) | 7 (24.1%) | |
| ICS/LABA | 13 (26%) | 14 (48.3%) | |
| LAMA/LABA | 0 (0%) | 3 (10.3%) | |

BMI, body mass index; LAMA, long-acting muscarinic antagonist; LABA, long-acting β-agonist; ICS, inhaled corticosteroids; SD, standard deviation; NS, not significant

TLA did not significantly change. The CSA<5 and %CSA<5 slightly but significantly increased in the COPD group (baseline vs two years later: 0.72 ± 0.19% vs 0.78 ± 0.21%, $P = 0.0074$). In contrast, the CSA<5 and %CSA<5 significantly decreased in the COPD with asthma-like features group (baseline vs two years later: 0.77 ± 0.22% vs 0.70 ± 0.21%, $P = 0.0148$). The annual changes in %CSA<5 were significantly different between the two groups (COPD vs asthma-like features: 0.051 ± 0.104 vs -0.035 ± 0.094, $P = 0.0007$).On the other hand, the mean WA% at the distal bronchi significantly decreased in the COPD group (baseline vs two years later: 83.9 ± 3.2% vs 82.0 ± 4.6%, $P = 0.0043$), while the WA% in the COPD with asthma-like features group did not significantly change during the follow-up period (baseline vs two years later: 83.9 ± 3.5% vs 83.2 ± 4.0%, not significant). The annual changes in mean fifth WA% were not significantly different between the two groups (COPD vs asthma-like features: -0.011 ± 0.025 vs -0.004 ± 0.021, $P = 0.1352$).

## Discussion

This is the first report of longitudinal morphological changes, including parameters of emphysema, airway wall thickness, and vascularity, in COPD with asthma-like features. The key findings of our study were as follows. First, emphysematous features (LAA%) significantly increased in both groups. Second, longitudinal changes in the percentage of small pulmonary vessels (%CSA<5) differed between COPD with asthma-like features and COPD alone. Among patients with COPD with asthma-like features, %CSA<5 significantly decreased

**Table 2. Longitudinal changes in CAT score, PFTs, and CT measurements.**

| | COPD Baseline Mean (±SD) | 2-Year Follow-up Mean (±SD) | *P* value (within group) | COPD with asthma-likefeatures Baseline Mean (±SD) | 2-Year Follow-up Mean (±SD) | *P* value (within group) |
|---|---|---|---|---|---|---|
| CAT score | 8.7 ± 6.5 | 8.6 ± 7.1 | NS | 9.4 ± 7.0 | 10.6 ± 7.4 | NS |
| FVC (L) | 3.15 ± 0.72 | 3.19 ± 0.77 | NS | 3.01 ± 0.91 | 2.96 ± 0.86 | NS |
| $FEV_1$ (L) | 1.83 ± 0.61 | 1.81 ± 0.62 | NS | 1.45 ± 0.55 | 1.41 ± 0.56 | NS |
| $FEV_1$/FVC (%) | 57.5 ± 11.8 | 55.9 ± 11.5 | 0.0381 | 48.2 ± 11.0 | 47.5 ± 11.7 | NS |
| $FEV_1$% predicted (%) | 67.9 ± 19.8 | 69.0 ± 21.3 | NS | 53.6 ± 17.7 | 53.9 ± 19.3 | NS |
| V50/V25 | 3.7 ± 1.2 | 4.1 ± 1.6 | 0.0217 | 3.3 ± 1.3 | 3.3 ± 1.5 | NS |
| TLA ($mm^2$) | 22272 ± 3091 | 22131 ± 3277 | NS | 22841 ± 4620 | 22823 ± 4824 | NS |
| LAA ($mm^2$) | 1648 ± 2709 | 2144 ± 3100 | < 0.0001 | 2787 ± 3606 | 3626 ± 4139 | < 0.0001 |
| LAA% (%) | 6.8 ± 10.4 | 8.8 ± 11.7 | < 0.0001 | 10.8 ± 13.0 | 14.1 ± 14.6 | < 0.0001 |
| CSA<5 ($mm^2$) | 157.0 ± 34.2 | 168.9 ± 39.7 | 0.0047 | 164.2 ± 33.4 | 147.7 ± 27.1 | 0.0034 |
| %CSA<5 (%) | 0.72 ± 0.19 | 0.78 ± 0.21 | 0.0074 | 0.77 ± 0.22 | 0.70 ± 0.21 | 0.0148 |
| B1 WA% | | | | | | |
| Third | 72.5 ± 7.2 | 72.0 ± 6.4 | NS | 71.0 ± 6.6 | 70.9 ± 9.0 | NS |
| Fifth | 83.9 ± 4.7 | 83.0 ± 4.6 | NS | 84.9 ± 4.1 | 83.6 ± 7.1 | NS |
| B10 WA% | | | | | | |
| Third | 71.4 ± 7.0 | 69.6 ± 6.7 | NS | 70.7 ± 7.5 | 71.1 ± 6.8 | NS |
| Fifth | 83.8 ± 3.8 | 80.9 ± 6.2 | 0.0004 | 82.8 ± 4.8 | 82.5 ± 4.7 | NS |
| Mean WA% | | | | | | |
| Third | 72.2 ± 6.1 | 70.9 ± 4.8 | NS | 70.9 ± 5.6 | 71.2 ± 6.5 | NS |
| Fifth | 83.9 ± 3.2 | 82.0 ± 4.6 | 0.0043 | 83.9 ± 3.5 | 83.2 ± 4.0 | NS |

CAT score, COPD assessment test score; PFTs, pulmonary function tests; FVC, forced vital capacity; $FEV_1$, forced expiratory volume in 1 second; V50/V25, forced expiratory flow at 50% vital capacity and forced expiratory flow at 25% vital capacity; TLA, total lung area; LAA, low-attenuation area; CSA<5, the cross-sectional area of pulmonary vessels <5 $mm^2$; WA%, wall area percentage; SD, standard deviation; NS, not significant

**Table 3. Annual changes in CAT score, PFTs, and CT measurements: Intra- and inter-group comparisons between patients with COPD with and without asthma-like features.**

| | COPD | | COPD with asthma-like features | | *P* value (between groups) |
|---|---|---|---|---|---|
| | **Value** | ***P* value (within group)** | **Value** | ***P* value (within group)** | |
| CAT score | -0.1 ± 3.0 | NS | 0.3 ± 1.8 | NS | NS |
| $FEV_1$ (L) | -0.01 ± 0.11 | NS | -0.02 ± 0.08 | NS | NS |
| $FEV_1$% predicted (%) | 0.010 ± 0.086 | NS | -0.001 ± 0.054 | NS | NS |
| LAA% (%) | 0.83 ± 1.72 | < 0.0001 | 0.40 ± 0.53 | < 0.0001 | NS |
| %CSA<5 (%) | 0.051 ± 0.104 | 0.0074 | -0.035 ± 0.094 | 0.0148 | 0.0007 |
| Mean WA% (%) | | | | | |
| Third | -0.007 ± 0.037 | NS | 0.002 ± 0.027 | NS | NS |
| Fifth | -0.011 ± 0.025 | 0.0043 | -0.004 ± 0.021 | NS | NS |

CAT score, COPD assessment test score; PFTs, pulmonary function tests; FVC, forced vital capacity; $FEV_1$, forced expiratory volume in 1 second; LAA, low-attenuation area; CSA<5, cross-sectional area of pulmonary vessels <5 $mm^2$; WA%, wall area percentage; NS, not significant.

The data are expressed as mean ± standard deviation.

The difference in CAT score was calculated by subtracting the value measured after 2 years from that measured at enrollment.

The difference in $FEV_1$ (L) was calculated by subtracting the value measured after 2 years from that measured at enrollment.

The difference in $FEV_1$% predicted (%) and the CT parameters (%) were calculated as follows: subtraction of the value measured after 2 years from that measured at enrollment, divided by the value at enrollment.

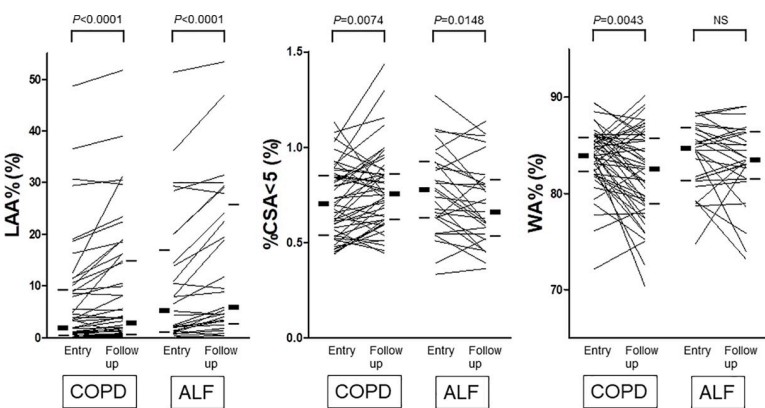

**Fig 2. Changes in radiological parameters over two years of follow-up.** Notes: Radiological parameters were compared between enrollment and two-year follow-up with the Wilcoxon signed-rank test. WA% shows the mean wall area percentage at the 5th generation of bronchi. The thick horizontal bars at the sides of each graph show the median, while the thin horizontal bars at the sides of each graph show the interquartile range. Abbreviations: ALF, asthma-like features; LAA, low-attenuation area; CSA<5, the cross-sectional area of pulmonary vessels < 5 $mm^2$; WA%, wall area percentage; NS, not significant.

during the two years of follow-up. In contrast, %CSA<5 significantly increased in patients with COPD alone. Third, we observed a slight but significant decrease in WA% in patients with COPD alone, although there was no significant change in WA% in patients with COPD with asthma-like features during the study period.

LAA% is recognized as an index of the extent of emphysema [12]. Emphysema is the main pathological lesion in COPD and is usually caused by tobacco smoke [1]. It is well known that LAA% gradually increases over time in COPD patients [30]. Patients with ACO have less severe emphysema than patients with COPD alone [25, 31]. Although there are a few reports on the cross-sectional structural characteristics of ACO [8, 32], longitudinal emphysematous changes in ACO have not been reported. Our findings suggest that emphysema progresses similarly in patients with COPD and those with COPD with asthma-like features. In COPD, exposure to tobacco smoke leads to chronic airway inflammation through the activation of inflammatory cells [33]. Regardless of smoking cessation, chronic inflammation persists [34] and causes emphysema progression [21]. In our study, most patients were past smokers, and all had a significant smoking history. Therefore, our data indicates that emphysematous destruction may progress in both patients with COPD and those with COPD with asthma-like features due to the fact that airway inflammation persist even after smoking cessation.

Another novel finding of this study was that changes in %CSA<5 differed between patients with COPD alone and those with asthma-like features. The %CSA<5 is a novel CT marker for microvascular alteration, and it is negatively correlated with the extent of emphysema and air-flow limitation in COPD [18]. However, Saruya et al. reported that the progression of emphysema and pulmonary vascular alteration do not always occur in parallel. They found that, although emphysema might progress, the %CSA<5 might not always decrease, and the change in decrease was associated with various patient improvements, such as smoking cessation, decreased exacerbation risk, and appropriate medical treatment [24]. We previously reported that %CSA<5 was higher in patients with ACO than in patients with COPD and inhaled corticosteroid (ICS)/long acting beta-agonist (LABA) treatment decreased %CSA<5 over 3 months in the ACO patients [35]. Vascular endothelial growth factor (VEGF) has an important role for the pathogenesis of several lung diseases [36]. In COPD, VEGF and VEGF receptor 2 are decreased, which causes the destruction of alveolar walls (emphysema) and loss of pulmonary microvessels [37]. On the contrary, airway inflammation occurs in response to increased

secretion of VEGF in asthma, which leads to abnormal vascularity and increased vessel size [38]. These difference in microvessels might lead to the difference in change of %CSA<5 we observed. Regarding the response to treatment, although previous studies reported that patients with ACO have a worse clinical course than patients with COPD alone [32, 39], recent studies have found that patients with ACO have a better prognosis [8, 9]. Suzuki et al. demonstrated that appropriate treatment improved the clinical course of patients with COPD with asthma-like features due to a better response to treatment [9]. Furthermore, Cosio et al. found that patients with ACO had a better 1-year prognosis than patients with COPD alone [8]. Therefore, various patient factors may affect the changes in %CSA<5 observed in both groups.

There was a significant decrease in WA% within patients with COPD alone, but with COPD with asthma-like features there was no significant change in WA% throughout the study period. Several transitional studies found that more airway wall thickening occurs in asthma than in COPD or controls [40, 41]. Patients with ACO had significantly greater wall thickening than those with COPD alone [25]. In addition, respiratory medications such as ICS and bronchodilators decreased airway wall thickening in both asthma and COPD [42, 43]. We previously demonstrated that the changes of airway wall thickness in patients with COPD alone may be associated to smoking cessation and appropriate treatment [26]. In this longitudinal study, the COPD patients with asthma-like features were more likely to have a severe GOLD stage at enrollment. Several previous reports also found that patients with ACO had lower $FEV_1$% at enrollment [39, 44]. We propose that more severe GOLD stage and greater airway remodeling were associated with the lesser change in WA% in patients with COPD with asthma-like features.

Another reason for no significant change in WA% among patients with COPD with asthma-like features may be that they had fewer features of asthma by our criteria compared to the criteria for ACO [5]. In the present study, 13 patients (48%) with COPD with asthma-like features had only one or two features of asthma. However, in a large cohort study (the COPD History Assessment in Spain (CHAIN) study), the diagnostic criteria required for ACO was one major criterion (bronchodilator test >400 mL and 15% and past medical history of asthma) or two minor criteria (blood eosinophils > 5%, IgE > 100 IU/mL, or two separate bronchodilator tests > 200 mL and 12%) [8]. We consider that the number of asthma-like features possibly may be associated with a decrease in WA% due to response to treatment.

Although there are few randomized clinical studies to provide guidance for the treatment of ACO, treatment with ICS is recommended for patients with ACO, like as for patients with asthma [5]. Recently, GOLD guideline recommends the treatment containing ICS for patients with high eosinophil counts in COPD [45]. In this study, the rate of ICS (including ICS/LABA) use was increased during the follow-up period. About 75 percent of patients with COPD with asthma-like features used ICS and ICS/LABA 2 years later. These findings support the strategy of considering treatment with ICS in patients with COPD with asthma-like features, even if they have not been already diagnosed with ACO [8, 9].

## Limitations

Our study has several limitations. First, the study only enrolled a small number of patients and was a preliminary and exploratory investigation conducted at a single institute. Second, the observational period of two years was relatively short. Third, because of the small number of subjects we could not evaluate each CT parameter for differences in treatment, with or without ICS, rate of smoking cessation, and frequency of exacerbations. Further prospective studies with larger study populations and longer observational period are required to confirm these results.

## Conclusions

In conclusion, emphysematous regions (LAA%) increased over time in patients with COPD with and without asthma-like features. The %CSA<5 and the WA% at the distal bronchi did not change in parallel with LAA in either group. Furthermore, the change in %CSA<5 over time in COPD with asthma- like features was significantly different than in COPD alone. Patients with COPD with asthma-like features may have different longitudinal structural changes compared to those observed in COPD patients.

## Supporting information

**S1 File. Participant raw data.**
(XLSX)

## Acknowledgments

We thank Dr. Toshio Suzuki, Dr. Akira Nishiyama, Mrs. Chieko Handa, Mrs. Tamie Hirano, and Miki Sakurai for their technical assistance and general support. We also wish to thank Dr. Shunsuke Furuta and Dr. Yuki Shiko for advice on the statistical analyses.

## Author Contributions

**Conceptualization:** Rie Anazawa, Naoko Kawata, Koichiro Tatsumi.

**Data curation:** Rie Anazawa, Naoko Kawata, Yukiko Matsuura, Shin Takayanagi.

**Formal analysis:** Rie Anazawa, Naoko Kawata, Yukiko Matsuura.

**Funding acquisition:** Naoko Kawata, Koichiro Tatsumi.

**Investigation:** Rie Anazawa, Naoko Kawata, Yukiko Matsuura, Shin Takayanagi.

**Methodology:** Rie Anazawa, Naoko Kawata, Yukiko Matsuura, Masaki Suzuki, Shin Takayanagi, Shin Matsuoka, Shoichiro Matsushita.

**Project administration:** Naoko Kawata, Koichiro Tatsumi.

**Resources:** Rie Anazawa, Naoko Kawata, Yukiko Matsuura, Jun Ikari, Yuji Tada.

**Software:** Rie Anazawa, Masaki Suzuki, Shin Matsuoka, Shoichiro Matsushita.

**Supervision:** Naoko Kawata, Koichiro Tatsumi.

**Validation:** Rie Anazawa, Naoko Kawata.

**Visualization:** Rie Anazawa, Naoko Kawata.

**Writing – original draft:** Rie Anazawa, Naoko Kawata.

**Writing – review & editing:** Rie Anazawa, Naoko Kawata, Jun Ikari, Yuji Tada, Shin Matsuoka, Koichiro Tatsumi.

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
