## [Decision Letter · Decision Letter 0]

11 Sep 2019

PONE-D-19-20122

Longitudinal changes in structural lung abnormalities using MDCT in chronic obstructive pulmonary disease with asthma-like features

PLOS ONE

Dear Dr Naoko Kawata,

Thank you for submitting your manuscript to PLOS ONE. After careful consideration, we feel that it has merit but does not fully meet PLOS ONE’s publication criteria as it currently stands. Therefore, we invite you to submit a revised version of the manuscript that addresses the points raised during the review process.

We would appreciate receiving your revised manuscript by Oct 26 2019 11:59PM. To enhance the reproducibility of your results, we recommend that if applicable you deposit your laboratory protocols in protocols.io, where a protocol can be assigned its own identifier (DOI) such that it can be cited independently in the future. For instructions see: http://journals.plos.org/plosone/s/submission-guidelines#loc-laboratory-protocols

We look forward to receiving your revised manuscript.

Kind regards,

Davor Plavec

Academic Editor

PLOS ONE

Journal Requirements:

Additional Editor Comments:

This is an interesting topic, especially because morphological features of the chronic progressive disease was followed. But there are some things that need to be improved prior to the decision about publication.

In the Methods section (MDCT) there is a need to clarify in more details how this was done: “All data were analyzed independently by two pulmonologists (RA and YM) who were blinded to all patient clinical information.” How the blinding was provided? Were they blinded also for the sequence (baseline and follow up)? How was the result calculated?

There is no data on sample size calculation and why this size of sample was used for this research. Please provide this! Also provide the Flowchart of the patients according to the STROBE! Please also provide the STROBE checklist!

In the Statistical analysis for the comparison of the follow up data an appropriate method of the analysis should be used – analysis of variance for repeated measurements. Please repeat your analyses using this method and report the results.

In Table 1 please express the rate of exacerbations per 1 year as this is the usual way to do it.

In Table 2 there is variable that is not explained anywhere – V50/V25. Please explain or remove if this is a mistake.

For Figure 1 there is no explanation on which data is represented on the box and whiskers plots! Please add this to your legend. Also it would be more appropriate if you would present the data for each variable for the comparing groups beside each other and not separately for each group. Please correct this. Also it would be better if you would present line graphs instead of box-plots.

In Discussion section there are too many speculations about possible mechanism like: “Indeed, interleukin (IL)-6 known as inflammatory biomarker was associated with emphysema progression (LAA) as assessed by CT scans in COPD [35]. Our data indicates that emphysematous destruction progresses in both patients with COPD and those with COPD with asthma-like features due to persistent airway inflammation even after smoking cessation.” or “We speculate that appropriate treatment, including ICS, decreases bronchial inflammation, which reduces vascular alterations near the small airways.”

Please correct this because these speculations could not be based or associated with you research results.

Please add the difference in management and other patient characteristics (smoking cessation rate) to the Limitations section of the manuscript.

Please change the last sentence in the Conclusions because it represents a too farfetched assumption.

Reviewers' comments:

Reviewer's Responses to Questions

**Comments to the Author**

1. Is the manuscript technically sound, and do the data support the conclusions?

Reviewer #1: Yes

2. Has the statistical analysis been performed appropriately and rigorously? 

Reviewer #1: I Don't Know

3. Have the authors made all data underlying the findings in their manuscript fully available?

Reviewer #1: Yes

4. Is the manuscript presented in an intelligible fashion and written in standard English?

Reviewer #1: Yes

5. Review Comments to the Author

Reviewer #1: I believe this is an important clue in the puzzle of COPD. This study should prompt more bigger studies to clarify the asthma COPD difference. I personally do not support the ACO but believe that patient can have both diseases in different ratios. Regards.

6. PLOS authors have the option to publish the peer review history of their article (what does this mean?). If published, this will include your full peer review and any attached files.

Reviewer #1: No

---

## [Author Response · Author response to Decision Letter 0]

15 Nov 2019

RESPONSE TO EDITOR:

EVALUATION: This is an interesting topic, especially because morphological features of the chronic progressive disease was followed. But there are some things that need to be improved prior to the decision about publication.

RESPONSE: We greatly appreciate the editor’s constructive comments, which have helped us to considerably improve the quality of our manuscript. We have revised our manuscript according to the editor’s comments. For easy reference, we have provided the revised text in red font.

COMMENT 1: In the Methods section (MDCT) there is a need to clarify in more details how this was done:“All data were analyzed independently by two pulmonologists (RA and YM) who were blinded to all patient clinical information.” How the blinding was provided? Were they blinded also for the sequence (baseline and follow up)? How was the result calculated?

RESPONSE: 

We appreciate your helpful comment. The MDCT measurements were evaluated by two pulmonologists (RA and YM) independently. All information including the MDCT data were anonymized and the observers were blinded to other characteristics of the subjects when imaging analysis were performed.

We have revised the manuscript and added the following sentences in the Materials and methods section regarding MDCT measurements (Page 5).

Page 5, lines 133-135, in Materials and methods

Before:

“All data were analyzed independently by two pulmonologists (RA and YM) who were blinded to all patient clinical information.”

After:

“The MDCT measurements were evaluated independently by two pulmonologists (RA and YM). All data were anonymized and the observers were blinded to other characteristics of the subjects when the imaging analyses were performed.”

COMMENT 2: There is no data on sample size calculation and why this size of sample was used for this research. Please provide this! Also provide the Flowchart of the patients according to the STROBE! Please also provide the STROBE checklist!

RESPONSE: We thank the editor for these important comments. We have rechecked the STROBE checklist 1 and have added a flow chart of the study participants as Figure1 below. 

We have also calculated the required sample size by JMP 13.0. The number of subjects needed for comparison of the two groups is 118 when the level of significance is 0.05, power (1-β) is 0.8, the difference is 0.1, and standard deviation is 0.2 2. However, the present study is an observational, preliminary, and exploratory study. A few previous studies on longitudinal structural changes have used these three CT parameters (LAA%, WA%, CSA%) mentioned in the present study in patients with COPD 2, 3. Tanabe et al. reported the impact of exacerbations on emphysema progression in COPD 4. They compared the change over two years in CT parameters related to emphysematous progression between the exacerbation group (n=34) vs a non-exacerbation group (n=26). In our previous study, we evaluated the change in three CT parameters in COPD (n=58) for two years 2 and described the morphological changes in the following groups: exacerbation (n=14) and non-exacerbation (n=44), and ex-smokers (n=46) and current smokers (n=12). 

 There are also a few studies on longitudinal structural changes using CT parameters in patients with ACO or those with COPD with asthma-like features. We have reported previously radiological findings between baseline and following budesonide/formoterol treatment in 20 patients with ACOS 3. Another report described CT parameters for emphysema and airway wall thickening between three groups: COPD group (n=118), ACOS group (n=32), and asthma with airflow limitation (n=27) 5. The present study had a small number of subjects with COPD with asthma-like features similar to these previous studies, and is therefore a preliminary and exploratory study. Accordingly, based on the sample size calculation and the advice of the statistician of our institution we consider it is better that the study is defined as a preliminary investigation. However, we would like to elucidate our findings in a further multi-center cohort study on a large number of subjects according to the editor’s very helpful insights.

We have revised the manuscript in the Limitation section.

Page 13, lines 299-300, in Limitation

Before:

“First, our sample size was small because this study was conducted at a single institute.”

After:

“First, the study only enrolled a small number of patients and was a preliminary and exploratory investigation conducted at a single institute”

We have also revised the manuscript and added the following text in the Materials and methods section regarding subjects (Page 3). We also added a flowchart of the patients as new Figure1 in the manuscript (Page 4).

Page 3, lines 74-81, in Materials and methods

“This prospective observational study enrolled 137 patients who presented to Chiba University Hospital from June 2010 to August 2016 for the management of COPD. Patients were excluded if they had abnormal lung parenchymal lesions other than emphysematous change. At enrollment, 10 patients were excluded for the following reasons: 4 interstitial pneumonia, 1 infectious pneumonia, 1 old pulmonary tuberculosis, and 4 lung cancer or nodules. At the follow up period, 15 patients were excluded: 6 interstitial pneumonia, 2 infectious pneumonia, 3 lung cancer or nodules, 2 cardiac failure, and 2 pleural effusion. Finally, 50 patients with COPD and 29 patients with COPD with asthma-like features were enrolled in the study (Figure 1).”

Fig 1. Flow chart of the study participants

COMMENT 3: In the Statistical analysis for the comparison of the follow up data an appropriate method of the analysis should be used – analysis of variance for repeated measurements. Please repeat your analyses using this method and report the results.

RESPONSE: We thank the editor for this comment. In accordance with the comment, we performed a repeated measure two factorial ANOVA. There was a significant relationship between the change in %CSA<5, and the interaction between COPD/ACO and elapsed time (P=0.003) 

The present study consisted of two groups and we measured each parameter two times. In previous reports using MDCT parameters 2, 4, the longitudinal changes within each two groups measured two times were evaluated by the Wilcoxon signed-rank test. For the longitudinal changes between the two groups, the difference between baseline and follow-up in each group was compared using the Mann-Whitney U test. 

We consulted regarding this point with the statistician at our institution. Finally, we decided to compare the longitudinal changes within each group in this preliminary study using the Wilcoxon signed-rank test. For longitudinal changes between the two groups, we compared differences in the changes using the Mann-Whitney U test. The results of the annual change in lung function and computed tomography parameters are shown in the new Table 3 as below.

Page 8-9, in Results

Table 3. Annual changes in CAT score, PFTs, and CT measurements: Intra- and inter-group comparisons between patients with COPD with and without asthma-like features

 COPD COPD with asthma-like features P value

(between groups)

 Value P value

(within group) Value P value

(within group) 

CAT score -0.1 ± 3.0 NS 0.3 ± 1.8 NS NS

FEV1 (L) -0.01 ± 0.11 NS -0.02 ± 0.08 NS NS

FEV1% predicted (%) 0.010 ± 0.086 NS -0.001 ± 0.054 NS NS

LAA% (%) 0.83 ± 1.72 < 0.0001 0.40 ± 0.53 < 0.0001 NS

%CSA<5 (%) 0.051 ± 0.104 0.0074 -0.035 ± 0.094 0.0148 0.0007

Mean　WA% (%) 

Third -0.007 ± 0.037 NS 0.002 ± 0.027 NS NS

Fifth -0.011 ± 0.025 0.0043 -0.004 ± 0.021 NS NS

CAT score, COPD assessment test score; PFTs, pulmonary function tests; FVC, forced vital capacity; FEV1, forced expiratory volume in 1 second; LAA, low-attenuation area; CSA<5, cross-sectional area of pulmonary vessels <5 mm2; WA%, wall area percentage; NS, not significant.

The data are expressed as mean ± standard deviation.

The difference in CAT score was calculated by subtracting the value measured after 2 years from that measured at enrollment.

The difference in FEV1 (L) was calculated by subtracting the value measured after 2 years from that measured at enrollment.

The difference in FEV1% predicted (%) and the CT parameters (%) were calculated as follows: subtraction of the value measured after 2 years from that measured at enrollment, divided by the value at enrollment.

We have revised and added the following sentences to the Results section regarding CT parameters (Page 10).

Page 10, lines 210-217, CT parameters in Results

“The annual changes in %CSA<5 were significantly different between the two groups (COPD vs asthma-like features: 0.051 ± 0.104 vs. -0.035 ± 0.094, P=0.0007). On the other hand, the mean WA% at the distal bronchi significantly decreased in the COPD group (baseline vs two years later: 83.9 ± 3.2% vs 82.0 ± 4.6%, P = 0.0043), while the WA% in the COPD with asthma-like features group did not significantly change during the follow-up period (baseline vs two years later: 83.9 ± 3.5% vs 83.2 ± 4.0%, not significant). The annual changes in mean fifth WA% were not significantly different between the two groups (COPD vs asthma-like features: -0.011 ± 0.025 vs -0.004 ± 0.021, not significant).”

We have revised the manuscript and added the following text in the Materials and methods section regarding the statistical analysis (Page 5-6). 

Page 5, lines 138-141, the statistical analysis in the Materials and methods

“JMP 13.0 software was used for all statistical analyses (SAS Institute, Cary, NC), with the results expressed as mean (± standard deviation) unless otherwise indicated. Comparison of longitudinal changes in CT parameters, PFTs, and CAT scores between enrollment and follow-up in each group was performed using the Wilcoxon signed-rank test. Comparisons were performed with a chi-square test for categorical variables. Differences in the data between the patients with COPD or COPD with asthma-like features were analyzed using the Mann-Whitney U test.”

COMMENT 4: In Table 1 please express the rate of exacerbations per 1 year as this is the usual way to do it.

RESPONSE: We have revised this in accordance with the editor's comment (Table 1, Page 7).

Page 7, in Table 1

 COPD COPD with asthma-like features P value

Exacerbations per 1 year 0.8 ± 0.5 1.6 ± 1.9 NS

COMMENT 5: In Table 2 there is variable that is not explained anywhere – V50/V25. Please explain or remove if this is a mistake.

RESPONSE: We appreciate your helpful comment. We have revised the table in accordance with the comment (Page 8).

Page 8, lines 178-179, in abbreviations of Table 2

“V50/V25, forced expiratory flow at 50% vital capacity and forced expiratory flow at 25% vital capacity”

COMMENT 6: For Figure 1 there is no explanation on which data is represented on the box and whiskers plots! Please add this to your legend. Also it would be more appropriate if you would present the data for each variable for the comparing groups beside each other and not separately for each group. Please correct this. Also it would be better if you would present line graphs instead of box-plots.

RESPONSE: We appreciate the reviewer’s helpful comment. We have revised the figure as a new Figure 2 in accordance with the comment). We have shown comparisons of the data for each variable in the groups beside each other, instead of separately using line graphs. We have also added an explanation of the CT parameters in the Figure legend in the Results section.

Fig 2. Changes in radiological parameters over two years of follow-up

Before:

After:

Page 9, line 219-223, in Results section 

Notes: Radiological parameters were compared between enrollment and two-year follow-up using the Wilcoxon signed-rank test. WA% shows the mean wall area percentage at the 5th generation of bronchi.

Abbreviations: ALF, asthma-like features; LAA, low-attenuation area; CSA<5, the cross-sectional area of pulmonary vessels < 5 mm2; WA%, wall area percentage; NS, not significant

COMMENT 7: In Discussion section there are too many speculations about possible mechanism like: “Indeed, interleukin (IL)-6 known as inflammatory biomarker was associated with emphysema progression (LAA) as assessed by CT scans in COPD [35]. Our data indicates that emphysematous destruction progresses in both patients with COPD and those with COPD with asthma-like features due to persistent airway inflammation even after smoking cessation.” or “We speculate that appropriate treatment, including ICS, decreases bronchial inflammation, which reduces vascular alterations near the small airways.”

Please correct this because these speculations could not be based or associated with you research results.

Please add the difference in management and other patient characteristics (smoking cessation rate) to the Limitations section of the manuscript.

Please change the last sentence in the Conclusions because it represents a too farfetched assumption.

RESPONSE: 

We appreciate this important comment. We have revised and added the following text in the Abstract, Discussion, and Conclusion section.

Page 1, lines 21-22, Conclusion in Abstract

Before:

“Furthermore, changes in %CSA<5 were significantly different between patients with COPD and those with COPD with asthma-like features. Appropriate medical management may have a different effect on structural changes in COPD with and without asthma-like features.”

After:

“Furthermore, changes in %CSA<5 were significantly different between patients with COPD and those with COPD with asthma-like features. Patients with COPD with asthma-like features may have different longitudinal structural changes than those seen in COPD patients.”

We revised the Discussion.

Page 11, lines 241-246, in Discussion

Before:

“Exposure to tobacco smoke leads to chronic airway inflammation through the activation of inflammatory cells [33]. Regardless of smoking cessation, chronic inflammation persists [34] and causes emphysema progression [21]. In our study, most patients were past smokers, and all had a significant smoking history. Indeed, interleukin (IL)-6 known as inflammatory biomarker was associated with emphysema progression (LAA) as assessed by CT scans in COPD [35]. Our data indicates that emphysematous destruction progresses in both patients with COPD and those with COPD with asthma-like features due to persistent airway inflammation even after smoking cessation.”

After:

“In COPD, exposure to tobacco smoke leads to chronic airway inflammation through the activation of inflammatory cells [33]. Regardless of smoking cessation, chronic inflammation persists [34] and causes emphysema progression [21]. Therefore, our data indicates that emphysematous destruction may progress in both patients with COPD and those with COPD with asthma-like features due to the fact that airway inflammation persist even after smoking cessation.”

Page 11-12, lines 261-268, in Discussion

Before:

“Moreover, appropriate treatment might reduce the %CSA<5 in the COPD patients with asthma-like features. Although previous studies reported that patients with ACO have a worse clinical course than patients with COPD alone [32, 40], recent studies have found that patients with ACO have a better prognosis [8, 9]. Suzuki et al. demonstrated that appropriate treatment improved the clinical course of patients with COPD with asthma-like features due to a better response to treatment [9]. Furthermore, Cosio et al. found that patients with ACO had a better 1-year prognosis than patients with COPD alone [8].”

After:

“Regarding the response to treatment, although previous studies reported that patients with ACO have a worse clinical course than patients with COPD alone [32, 39], recent studies have found that patients with ACO have a better prognosis [8, 9]. Suzuki et al. demonstrated that appropriate treatment improved the clinical course of patients with COPD with asthma-like features due to a better response to treatment [9]. Furthermore, Cosio et al. found that patients with ACO had a better 1-year prognosis than patients with COPD alone [8]. Therefore, these various patient factors may affect the changes in %CSA<5 observed in both groups.”

Page 12-13, lines 294-296, in Discussion

Before:

“Optimal management including ICS reduces the airway inflammation and may provide a greater effect on patients with COPD with asthma-like features.”

After:

“These findings support the strategy of considering treatment with ICS in patients with COPD with asthma-like features, even if they have not been already diagnosed with ACO [8, 9].”

We have also revised and added differences in management and other patient characteristics (smoking cessation rate) to the Limitations section of the manuscript.

Page 13, lines 301-304, in Limitation

“Third, because of the small number of subjects we could not evaluate each CT parameter for differences in treatment, with or without ICS, rate of smoking cessation, and frequency of exacerbations. Further prospective studies with larger study populations and longer observational period are required to confirm these results.”

The Conclusion has been revised as follows.

Page 13, lines 310-312, in Conclusion

Before:

“Management strategies may have different effects on pulmonary vascularity between COPD with and without asthma-like features.”

After:

“Patients with COPD with asthma-like features may have different longitudinal structural changes compared to those observed in COPD patients.

 

RESPONSE TO REVIEWER:

EVALUATION: I believe this is an important clue in the puzzle of COPD. This study should prompt more bigger studies to clarify the asthma COPD difference. I personally do not support the ACO but believe that patient can have both diseases in different ratios. Regards.

RESPONSE: 

We greatly appreciate your helpful and important insights on the subjects with asthma-COPD overlap (ACO). As the reviewer has commented, the pathogenesis and physiological findings are distinct between COPD and asthma 6. Recently, ACO has been recognized in clinical practice. Although GINA/GOLD published a consensus on ACO 7, the diagnosis of ACO has not been defined definitively, and previous studies have used multiple different diagnostic criteria 8. On the other hand, some patients with COPD have asthma-like features 9, even if they have not been diagnosed with ACO. Several previous reports have described the characteristics of ACO and COPD with asthma-like features 3, 9, 10. COPD with asthma-like features might be a clinical phenotype of COPD, and identifying this phenotype is used for designing individualized treatments 9, 11, 12. We consider that larger studies on COPD, asthma, and ACO are required in the near future in order to elucidate the characteristics of each phenotype 13, 14.

 

References

1. von Elm E, Altman DG, Egger M, et al. The Strengthening the Reporting of Observational Studies in Epidemiology (STROBE) Statement: guidelines for reporting observational studies. Int J Surg. 2014; 12: 1495-9.

2. Takayanagi S, Kawata N, Tada Y, et al. Longitudinal changes in structural abnormalities using MDCT in COPD: do the CT measurements of airway wall thickness and small pulmonary vessels change in parallel with emphysematous progression? Int J Chron Obstruct Pulmon Dis. 2017; 12: 551-60.

3. Suzuki T, Tada Y, Kawata N, et al. Clinical, physiological, and radiological features of asthma-chronic obstructive pulmonary disease overlap syndrome. Int J Chron Obstruct Pulmon Dis. 2015; 10: 947-54.

4. Tanabe N, Muro S, Hirai T, et al. Impact of exacerbations on emphysema progression in chronic obstructive pulmonary disease. Am J Respir Crit Care Med. 2011; 183: 1653-9.

5. Kitaguchi Y, Yasuo M and Hanaoka M. Comparison of pulmonary function in patients with COPD, asthma-COPD overlap syndrome, and asthma with airflow limitation. Int J Chron Obstruct Pulmon Dis. 2016; 11: 991-7.

6. Barnes PJ. Against the Dutch hypothesis: asthma and chronic obstructive pulmonary disease are distinct diseases. Am J Respir Crit Care Med. 2006; 174: 240-3; discussion 3-4.

7. Asthma COPD and Ashtma-COPD Overlap Syndrome (ACOS). Global Initiative for Asthma and Global Initiative for Chronic Obstructive Lung Disease 2015. Available from: https://goldcopd.org/asthma-copd-asthma-copd-overlap-syndrome/.

8. Tho NV, Park HY and Nakano Y. Asthma-COPD overlap syndrome (ACOS): A diagnostic challenge. Respirology. 2016; 21: 410-8.

9. Suzuki M, Makita H, Konno S, et al. Asthma-like Features and Clinical Course of Chronic Obstructive Pulmonary Disease. An Analysis from the Hokkaido COPD Cohort Study. Am J Respir Crit Care Med. 2016; 194: 1358-65.

10. Hardin M, Cho M, McDonald ML, et al. The clinical and genetic features of COPD-asthma overlap syndrome. Eur Respir J. 2014; 44: 341-50.

11. Leigh R, Pizzichini MM, Morris MM, Maltais F, Hargreave FE and Pizzichini E. Stable COPD: predicting benefit from high-dose inhaled corticosteroid treatment. Eur Respir J. 2006; 27: 964-71.

12. Fattahi F, ten Hacken NH, Lofdahl CG, et al. Atopy is a risk factor for respiratory symptoms in COPD patients: results from the EUROSCOP study. Respir Res. 2013; 14: 10.

13. Hizawa N. LAMA/LABA vs ICS/LABA in the treatment of COPD in Japan based on the disease phenotypes. Int J Chron Obstruct Pulmon Dis. 2015; 10: 1093-102.

14. Agusti A, Gea J and Faner R. Biomarkers, the control panel and personalized COPD medicine. Respirology. 2016; 21: 24-33.

---

## [Editor Report · Decision Letter 1]

25 Nov 2019

PONE-D-19-20122R1

Longitudinal changes in structural lung abnormalities using MDCT in chronic obstructive pulmonary disease with asthma-like features

PLOS ONE

Dear Dr Naoko Kawata,

Thank you for submitting your manuscript to PLOS ONE. After careful consideration, we feel that it has merit but does not fully meet PLOS ONE’s publication criteria as it currently stands. Therefore, we invite you to submit a revised version of the manuscript that addresses the points raised during the review process.

We would appreciate receiving your revised manuscript by Jan 09 2020 11:59PM. To enhance the reproducibility of your results, we recommend that if applicable you deposit your laboratory protocols in protocols.io, where a protocol can be assigned its own identifier (DOI) such that it can be cited independently in the future. For instructions see: http://journals.plos.org/plosone/s/submission-guidelines#loc-laboratory-protocols

We look forward to receiving your revised manuscript.

Kind regards,

Davor Plavec

Academic Editor

PLOS ONE

Additional Editor Comments (if provided):

Dear Authors,

your manuscript is acceptable for publication after small revisions that should be done on Figures. On Figure 1 please change the words "abnormal chest shadows" into something more appropriate, like "abnormal lung parenchymal lesions other than emphysematous change". On Figure 2, and also the markings for the median and interquartile range for each time point and group.

---

## [Author Response · Author response to Decision Letter 1]

11 Dec 2019

RESPONSE TO EDITOR:

EVALUATION:

Your manuscript is acceptable for publication after small revisions that should be done on Figures. On Figure 1 please change the words "abnormal chest shadows" into something more appropriate, like "abnormal lung parenchymal lesions other than emphysematous change". On Figure 2, and also the markings for the median and interquartile range for each time point and group.

RESPONSE:

We greatly appreciate the editor’s constructive comments which have helped us to considerably improve the quality of our manuscript. We have revised our manuscript according to the editor’s comments.

1) We revised Figure 1 according to the editor’s comment.

Fig 1. Flow chart of the study participants

2) We also revised Figure 2 according to the editor’s comment. We have added bars at which represent the median and interquartile range (Page 11).

Fig 2. Changes in radiological parameters over two years of follow-up

Page 11, lines 220-223, in Results

Before:

“Notes: Radiological parameters were compared between enrollment and two-year follow-up with the Wilcoxon signed-rank test. WA% shows the mean wall area percentage at the 5th generation of bronchi.”

After:

“Notes: Radiological parameters were compared between enrollment and two-year follow-up with the Wilcoxon signed-rank test. WA% shows the mean wall area percentage at the 5th generation of bronchi. The thick horizontal bars at the sides of each graph show the median, while the thin horizontal bars at the sides of each graph show the interquartile range.”

---

## [Editor Report · Decision Letter 2]

13 Dec 2019

Longitudinal changes in structural lung abnormalities using MDCT in chronic obstructive pulmonary disease with asthma-like features

PONE-D-19-20122R2

Dear Dr. Naoko Kawata,

We are pleased to inform you that your manuscript has been judged scientifically suitable for publication and will be formally accepted for publication once it complies with all outstanding technical requirements.

With kind regards,

Davor Plavec

Academic Editor

PLOS ONE

Additional Editor Comments (optional):

Dear Authors,

after you have made the requested changes your manuscript is acceptable in this form for publication in PLOS ONE.
---

## [Editor Report · Acceptance letter]

18 Dec 2019

PONE-D-19-20122R2 

Longitudinal changes in structural lung abnormalities using MDCT in chronic obstructive pulmonary disease with asthma-like features 

Dear Dr. Kawata:

I am pleased to inform you that your manuscript has been deemed suitable for publication in PLOS ONE. Congratulations! Your manuscript is now with our production department. 

With kind regards,

on behalf of

Dr. Davor Plavec 

Academic Editor

PLOS ONE